# Rethinking Supervised Fine-Tuning: Emphasizing Key Answer Tokens for Improved LLM Accuracy

## Abstract

With the rapid advancement of Large Language Models (LLMs), the Chain-of-Thought (CoT) component has become significant for complex reasoning tasks. However, in conventional Supervised Fine-Tuning (SFT), the model could allocate disproportionately more attention to CoT sequences with excessive length. This reduces focus on the much shorter but essential Key portion-the final answer, whose correctness directly determines task success and evaluation quality. To address this limitation, we propose SFTKey, a two-stage training scheme. In the first stage, conventional SFT is applied to ensure proper output format, while in the second stage, only the Key portion is fine-tuned to improve accuracy. Extensive experiments across multiple benchmarks and model families demonstrate that SFTKey achieves an average accuracy improvement exceeding 5% over conventional SFT, while preserving the ability to generate correct formats. Overall, this study advances LLM fine-tuning by explicitly balancing CoT learning with additional optimization on answer-relevant tokens.

## 1 Introduction

Large Language Models (LLMs) with billions of parameters have achieved remarkable performance across a wide range of complex language tasks (Team, 2025; Guo et al., 2025; OpenAI, 2024). Typically built on Transformer architectures and trained in unsupervised pretraining followed by Supervised Fine-Tuning(SFT) on labeled prompt–response pairs, LLMs are capable of instruction following, complex reasoning, and generating desired outputs (Radford et al., 2019; Zhou et al., 2023; Li et al., 2023). The SFT stage shifts the model's objective from next-token prediction towards instruction following and answer generation, adapting the pretrained model to domain specific knowledge and scenarios. Studies(Zhou et al., 2023; Kirstain et al., 2021) show that even with relatively small datasets, SFT can yield substantial performance gains on downstream tasks, strengthening instruction following ability and output consistency.

In particular, many synthetic datasets generated for SFT consist of carefully curated Chain-of-Thought (CoT) segments of intermediate reasoning steps followed by a concise final answer(Wei et al., 2023; Cobbe et al., 2021; Mihaylov et al., 2018). The long CoT reasoning texts bridge the gap between the question prompt and the final answer, aligning the model's inference with human-like cognitive processes and improving its capability of complex reasoning. Conventional paradigm for SFT treats each token in the target response equally, minimizing the negative log-likelihood over the entire sequence. This uniform optimization, however, may risk overfitting reasoning tokens while neglecting the Key portion – the final answer segment that ultimately determines task success.

Several recent works study the non-uniform token weighting methods during fine-tuning. For instance, SFT-GO(Kim et al., 2025) groups tokens by importance (e.g. via TF–IDF) and optimizes a worst-group loss so that informative token groups are fully explored. Similarly, the Forgetting framework(Ghahrizjani et al., 2025) explicitly classifies tokens as positive or negative based on their utility and then down-weights the less useful tokens during fine-tuning. These approaches affirm the intuition that not all tokens contribute equally to the model performance. However, they introduce extra hyperparameters or judge models for the token selection, which not only increases the complexity of the training process but also requires carefully tuning to balance the contribution of

various token groups effectively. Another line of work aims to shorten reasoning chains or compress inputs to improve efficiency, such as prompt compression (Xia et al., 2025; Jiang et al., 2023) and long–short chain mixture SFT (Yu et al., 2025). These approaches show that redundant tokens can be safely removed to improve efficiency without degrading accuracy. Nevertheless, they typically rely on training or designing an additional rewrite model, which also introduces extra computation and may face generalization challenges when applied to unseen domains or reasoning styles. Therefore, there remains a need for a simple, general mechanism to improve the answer accuracy while preserving its reasoning ability.

In this work, we first propose a new two-stage training scheme called **SFTKey**. In the first stage, we apply standard SFT to ensure correct output format. Next, we fine-tune the model only on the Key tokens to improve accuracy, which represent the final answer. In order to clearly distinguish reasoning text from the answer, we insert the special symbols `<Thinking></Thinking>` and `<Answer></Answer>` into the data referred as Tag. Based on SFTKey, we further introduce **SFTKey-Tag** method which trains LLMs on the data with Tag. This design helps the model focus on the key portion and alleviates the imbalance caused by long CoT sequences. Experiments on multiple benchmarks demonstrate that SFTKey-Tag outperforms standard SFT and other variants in accuracy, while maintaining complete reasoning and correct output format.

Our contributions are summarized as follows:

- We propose SFTKey, a simple two-stage SFT framework that explicitly boosts the importance of answer tokens while preserving reasoning context.
- We conduct a systematic analysis of the impact of Tag on model accuracy, highlighting its variability across different scenarios.
- We perform a comprehensive comparison between SFT-Tag and Key-Tag training strategies, elucidating their respective strengths and limitations, which motivates our design of a multi-stage optimization approach **SFTKey-Tag**.

## 2 METHODOLOGY

### 2.1 STANDARD SUPERVISED FINE-TUNING (SFT)

In conventional supervised fine-tuning for language models, we assume a dataset of $N$ prompt-response pairs $\mathcal{D} = \{(\boldsymbol{x}_i, \boldsymbol{y}_i)\}_{i=1}^{N}$, where each $\boldsymbol{x}_i$ denotes a prompt sequence and $\boldsymbol{y}_i$ denotes the corresponding desired response. The language model, parameterized by $\boldsymbol{\theta}$, is trained to minimize the negative conditional log-likelihood over the entire response:

$$\mathcal{L}_{\text{SFT}}(\boldsymbol{\theta}) = -\sum_{i=1}^{N} \sum_{t=1}^{L_i} \log P\left(y_{i,t} \mid \boldsymbol{x}_i, \boldsymbol{y}_{i,<t}; \boldsymbol{\theta}\right), \tag{1}$$

where $L_i$ is the length of the response sequence $\boldsymbol{y}_i$, and $\boldsymbol{y}_{i,<t}$ represents all tokens preceding position $t$ in the target. This objective results in updates to all parameters $\boldsymbol{\theta}$ based on the entire sequence.

### 2.2 SFTKEY TAG

We follow the structured training approach(Guo et al., 2025) that explicitly decomposes each response sequence in the training corpus into two distinct segments: a reasoning segment (denoted by the special tokens `<Thinking>` and `</Thinking>`) and an answer segment (denoted by `<Answer>` and `</Answer>`). Formally, each response $\boldsymbol{y}_i$ is reconstructed as:

$$\hat{\boldsymbol{y}}_i = \left[\texttt{<Thinking>}\boldsymbol{y}_i^{(\text{think})}\texttt{</Thinking>}\texttt{<Answer>}\boldsymbol{y}_i^{(\text{answer})}\texttt{</Answer>}\right], \tag{2}$$

The special tokens themselves are included in the sequence and are seen by the model. Our method consists of two sequential training stages, designed to first learn a general response capability and then refine the final answer generation.

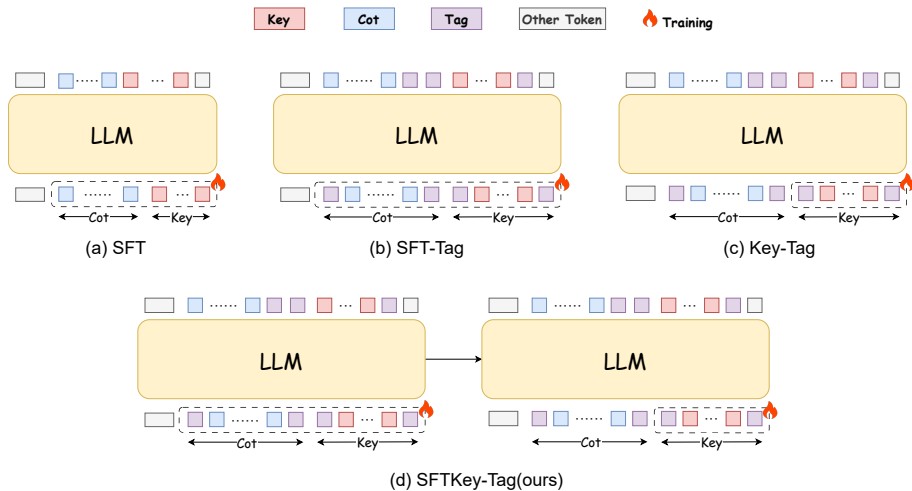

Figure 1: As illustrated in the figure, we compare four training strategies: SFT, SFT-Tag, Key-Tag, and SFTKey-Tag. In our setup, the training data are divided into two parts: the chain-of-thought (CoT) and the key (final answer). Building upon the baseline SFT, we further examine three variants: (i) SFT-Tag, which highlights the answer portion using special tags, (ii) Key-Tag, which trains exclusively on the key part, and SFTKey-Tag, a two-stage approach that combines the strategies of (i) SFT-Tag and (ii) Key-Tag.

**Stage 1: Tag enhanced SFT**

The model is first trained on the entire dataset $\hat{\mathcal{D}} = \{(\boldsymbol{x}_i, \hat{\boldsymbol{y}}_i)\}_{i=1}^{N}$ using the standard SFT objective (Eq. 1), which contains both the special token segments:

$$\boldsymbol{\theta}_{\text{SFT}} = \arg\min_{\boldsymbol{\theta}} \mathcal{L}_{\text{SFT}}(\boldsymbol{\theta}). \tag{3}$$

This stage ensures the model learns a robust policy for generating reasoning steps followed by answers, providing a base initialization $\boldsymbol{\theta}_{\text{SFT}}$.

**Stage 2: Key Specialized Fine-Tuning**

After convergence in Stage 1, we continue training the model, but now only the tokens within the `<Answer>` segment contribute to the loss and parameter updates. The tokens in the `<Thinking>` segment are still provided as context to the model but are excluded from gradient computation.

For each example $(\boldsymbol{x}_i, \hat{\boldsymbol{y}}_i)$, let $T_i$ be the index of the final token within the `</Think>` tag, marking the end of the reasoning segment. The loss function for the second stage is defined as:

$$\mathcal{L}_{\text{Answer}}(\boldsymbol{\theta}) = -\sum_{i=1}^{N} \sum_{t=T_i+1}^{L_i} \log P\left(y_{i,t} \mid \boldsymbol{x}_i, \hat{\boldsymbol{y}}_{i,<t}; \boldsymbol{\theta}\right). \tag{4}$$

The parameters are then updated by minimizing this specialized loss, initializing from the parameters obtained in Stage 1:

$$\boldsymbol{\theta}_{\text{SFT-Key}} = \arg\min_{\boldsymbol{\theta}} \mathcal{L}_{\text{Answer}}(\boldsymbol{\theta}), \quad \text{where } \boldsymbol{\theta} \text{ is initialized from } \boldsymbol{\theta}_{\text{SFT}}. \tag{5}$$

## 3 EXPERIMENTAL SETUP

**Models and Datasets**

To comprehensively evaluate the effectiveness of different training strategies, we conduct experiments on five representative language models—Qwen2.5-7B, Qwen2.5-3B, Qwen2.5-1.5B (Team,

2024), Qwen3-8B-Base (Team, 2025), and SmolLM3-3B-Base (Bakouch et al., 2025)—covering different model architectures and parameter scales. For evaluation, we employ four benchmark datasets spanning multiple reasoning and knowledge-intensive tasks: GSM8K (Cobbe et al., 2021), a widely used arithmetic reasoning benchmark consisting of 8.5K grade-school math word problems with detailed solutions; OpenR1-Math-220K[1], a large-scale dataset of 220K problems covering algebra, geometry, probability, and more, providing broader mathematical evaluation than GSM8K; OpenBookQA (Mihaylov et al., 2018), a multiple-choice QA benchmark designed to test scientific reasoning by integrating open-domain knowledge with reasoning skills; and CoT-Collection (Kim et al., 2023), a curated corpus of chain-of-thought annotated problems across domains, explicitly assessing reasoning trace quality beyond final-answer accuracy. The inclusion of heterogeneous datasets and models with different parameter scales ensures a wide-ranging and comprehensive evaluation of both generalization and reasoning effectiveness.

**Data Preprocessing** To enable explicit separation between reasoning and final answers, we introduce structural tokens `<Thinking></Thinking>` and `<Answer></Answer>`. Each dataset is manually divided into reasoning and answer segments, which are then recombined into the format `<Thinking>Thinking Tokens</Thinking><Answer>Answer Tokens</Answer>`. This strategy highlights the answer span while maintaining consistency with the original data distribution. We refer to this structural-token-based approach as Tag.

**Training Strategies** We explore four training strategies:
• **SFT** (Baseline Supervised Fine-Tuning): A conventional supervised fine-tuning approach conducted on the full training data without introducing any additional structural tokens. This serves as the baseline method.
• **SFT-Tag** (Tag Enhanced SFT): Supervised fine-tuning augmented with Tag that explicitly demarcate reasoning and answer segments. Both segments are included in optimization, enabling the model to better distinguish between intermediate reasoning and final outputs.
• **Key-Tag** (Answer-Focused Fine-Tuning with Tag): Fine-tuning performed exclusively on the answer segments. Data is still reformatted into the Tag structure , but during training, only the tokens within the `<Answer>` span contribute to the loss. This approach enforces stronger supervision on final outputs while retaining structural consistency.
• **SFTKey-Tag** (Two-Stage Fine-Tuning with Tag):
A two-stage training stage. In the first stage, standard SFT is applied to the full data. In the second stage, fine-tuning is restricted to the answer segments, similar to Key-Tag. Throughout both stages, the Tag format with Tag is consistently used to maintain explicit structural separation.

**Training Details** The learning rate is set to $5 \times 10^{-6}$ with a linear warmup of 0.5 epochs, and a weight decay of 0.1 is applied to improve generalization. Training is conducted using mixed precision with bfloat16, enhancing both computational efficiency and numerical stability. The number of training epochs is adjusted according to dataset size, typically ranging from 3 epochs. Similarly, We set the per-device batch size to 32. This keeps the effective batch size reasonable. All methods use the same optimization settings for fair comparison. Detailed GPU configurations are in subsection C.1. Additional training details are in Appendix B.

**Evaluation Methodology** After the model generates both the chain-of-thought (CoT) and the key (final answer), we primarily evaluate the key using an answer-level strategy, where predictions are extracted and systematically compared against gold-standard references. To improve robustness and reduce potential ambiguity in matching, we leverage an external large-scale language model, Meta-Llama-3-70B-Instruct (AI@Meta, 2024), as a reference judge to verify whether the generated outputs semantically align with the correct answers. In addition, we evaluate the output format by performing structured matching to determine whether the responses adhere to the expected format. Further details of the evaluation protocol are provided in Appendix D.

---

[1]https://huggingface.co/datasets/open-r1/OpenR1-Math-220k

# 4 MAIN RESULTS

| Model | Method | GSM8K | OpenR1-Math-220k | OpenBookQA | CoT-Collection | Avg-Score |
|-------|--------|-------|------------------|------------|----------------|-----------|
| Qwen3-8B-Base | SFT(baseline) | 0.8218 | 0.6164 | 0.9020 | 0.7278 | 0.7670 |
| | SFT-Tag | 0.8805 | 0.6959 | **0.9062** | 0.7264 | 0.8022 |
| | Key-Tag | 0.7360 | 0.5831 | 0.8762 | 0.6921 | 0.7218 |
| | **SFTKey-Tag(ours)** | **0.8816** | **0.8633** | 0.9020 | **0.7298** | **0.8441(+10.05%)** |
| Qwen2.5-7B | SFT(baseline) | 0.8305 | 0.5772 | 0.8922 | 0.7348 | 0.7586 |
| | SFT-Tag | 0.8302 | 0.5604 | 0.8992 | **0.7370** | 0.7567 |
| | Key-Tag | 0.5519 | 0.3822 | 0.6272 | 0.4480 | 0.5023 |
| | **SFTKey-Tag(ours)** | **0.8420** | **0.7217** | **0.9214** | 0.7342 | **0.8048(+6.07%)** |
| SmolLM3-3B-Base | SFT(baseline) | 0.7685 | 0.4574 | **0.8516** | 0.6678 | 0.6863 |
| | SFT-Tag | 0.7775 | **0.5645** | 0.7134 | 0.6502 | 0.6764 |
| | Key-Tag | 0.5850 | 0.2660 | 0.6192 | 0.3694 | 0.4599 |
| | **SFTKey-Tag(ours)** | **0.7798** | 0.5225 | 0.8200 | **0.6797** | **0.7005(+2.06%)** |
| Qwen2.5-3B | SFT(baseline) | 0.4749 | 0.3094 | 0.5138 | 0.3724 | 0.4176 |
| | SFT-Tag | 0.3866 | 0.3402 | 0.4984 | 0.3752 | 0.4001 |
| | Key-Tag | **0.5030** | **0.357** | **0.553** | 0.343 | **0.4390** |
| | **SFTKey-Tag(ours)** | 0.4739 | 0.3416 | 0.497 | **0.3997** | 0.4280(+2.49%) |
| Qwen2.5-1.5B | SFT(baseline) | 0.3248 | 0.2842 | 0.4536 | 0.3248 | 0.3468 |
| | SFT-Tag | 0.3332 | 0.2800 | 0.5754 | 0.3332 | 0.3804 |
| | Key-Tag | 0.3430 | 0.2828 | **0.5866** | **0.3430** | **0.4517** |
| | **SFTKey-Tag(ours)** | **0.3688** | **0.3434** | 0.5208 | 0.3192 | 0.3880(+4.12%) |

Table 1: Performance of different models trained with various strategies across multiple datasets, where each value represents a composite score that integrates both accuracy and output format adherence. The last column reports the average composite score for each model under a given training strategy. For each dataset within the same model, the highest composite score is highlighted in bold, and relative improvements over SFT are shown in parentheses. Detailed results for accuracy and format adherence are provided in Appendix E and Appendix F, respectively.

When assessing performance, we find that the Key-Tag approach consistently delivers higher answer accuracy compared to other training strategies. However, this gain comes at the cost of reduced output format adherence, an issue that we analyze in detail in the subsequent ablation study(Section 5.2). To provide a more balanced evaluation that accounts for this trade-off, we introduce a composite score, denoted as Score, which integrates both accuracy and format consistency:

$$Acc = \frac{1}{N} \sum_{i=1}^{N} \mathbf{1}[M_{a_i} = T_{a_i}]$$

$$Fmt = \frac{1}{N} \sum_{i=1}^{N} \mathbf{1}[M_{f_i} = T_{f_i}]$$

$$Score = \alpha \cdot Acc + (1 - \alpha) \cdot Fmt$$

Where $[M_{a_i}$ and $T_{a_i}$ denote the model's predicted and correct answers, and $M_{f_i}$ and $T_{f_i}$ denote the predicted and correct structures for the $i$-th question, with $N$ being the total number of questions. The weights for accuracy and format quality are $\alpha$ and $1 - \alpha$, and we set $\alpha = 0.7$ in our experiments.

As shown in Table 1, SFTKey-Tag achieves the highest composite score across all three general-capability models (Qwen3-8B, Qwen2.5-7B, SmolLM3-3B), demonstrating its effectiveness in simultaneously enhancing answer accuracy and maintaining well-structured outputs. For smaller models (Qwen2.5-3B, Qwen2.5-1.5B), due to their limited base capabilities, the format scores tend to be lower; under the current composite metric, accuracy dominates the overall score, which allows the Key-Tag approach to show relatively strong performance. Nevertheless, compared with the baseline SFT, our proposed SFTKey-Tag consistently outperforms across all evaluated models and datasets, with an overall improvement of approximately 5%.

To further analyze the impact of model scale, we group the results into two categories based on parameter size: large models (7B and 8B) and smaller models (3B and 1.5B). From this categorization, we observe that the composite score improvements are more pronounced for the larger models, suggesting that models with greater capacity derive greater benefit from the SFTKey-Tag training strategy. This finding highlights the synergistic effect of model scale and answer-focused fine-tuning on overall performance.

## 5 ABLATION STUDY

### 5.1 IMPACT OF TAGGING STRATEGY ON MODEL PERFORMANCE

| Model | Method | GSM8K | OpenR1-Math-220k | OpenBookQA | CoT-Collection | Avg-Acc |
|---|---|---|---|---|---|---|
| Qwen3-8B-Base | SFT(baseline) | **0.8378** | 0.5180 | 0.8600 | **0.6120** | 0.7069 |
| | SFT-Tag | 0.8300 | **0.6134** | **0.8660** | 0.6092 | **0.7296** |
| Qwen2.5-7B | SFT(baseline) | **0.7589** | **0.4620** | 0.8460 | 0.6220 | **0.6722** |
| | SFT-Tag | 0.7582 | 0.4380 | **0.8560** | **0.6260** | 0.6696 |
| SmolLM3-3B-Base | SFT(baseline) | 0.6732 | 0.3140 | **0.7880** | **0.5280** | **0.5758** |
| | SFT-Tag | **0.6854** | **0.4194** | 0.6180 | 0.5030 | 0.5565 |
| Qwen2.5-3B | SFT(baseline) | **0.6785** | 0.4420 | **0.7340** | 0.5320 | **0.5966** |
| | SFT-Tag | 0.5524 | **0.4860** | 0.7120 | **0.5360** | 0.5716 |
| Qwen2.5-1.5B | SFT(baseline) | 0.4640 | **0.4060** | 0.6480 | 0.4640 | 0.4955 |
| | SFT-Tag | **0.4760** | 0.4000 | **0.8220** | **0.4760** | **0.5435** |

Table 2: Accuracy performance of different models trained with SFT (baseline) and SFT-Tag strategies across multiple datasets. For each dataset within the same model, the higher value between SFT and SFT-Tag is highlighted in bold. The last column reports the average accuracy for each model.

Motivated by the design of DeepSeekR1(Guo et al., 2025), we introduce Tag, namely `<Thinking></Thinking>` and `<Answer></Answer>`, into the training data while keeping the baseline SFT methodology. This allowed us to systematically investigate whether adding such tags could help guide the model toward higher accuracy. The results show that the effect of these Tag varies across datasets, and improvements in accuracy are not consistently observed. This indicates that the accuracy of SFT varies under the influence of tags. Detailed experimental results are summarized in Table 2.

### 5.2 EVALUATING SFT AND KEY UNDER IDENTICAL TAGGING

To further examine the effect of training strategies, we conducted evaluations on five models: Qwen2.5-7B, Qwen3-8B, SmolLM-3B, Qwen2.5-3B, and Qwen2.5-1.5B, with detailed results provided in the appendix D. This diverse set of models allows for a comprehensive investigation of the behavior of SFT-Tag and Key-Tag across different architectures and parameter scales. As shown in Figure 2, each radar chart illustrates the accuracy differences between Key-Tag and SFT-Tag for a given model across multiple datasets. In four out of five cases, Key-Tag delivers clear accuracy improvements, while the remaining model shows comparable performance. The aggregated box ("Avg-acc") summarizes the distribution of mean accuracy differences across all models, confirming that Key-Tag consistently improves answer accuracy by emphasizing training on answer tokens, while maintaining robust generalization across model families and sizes.

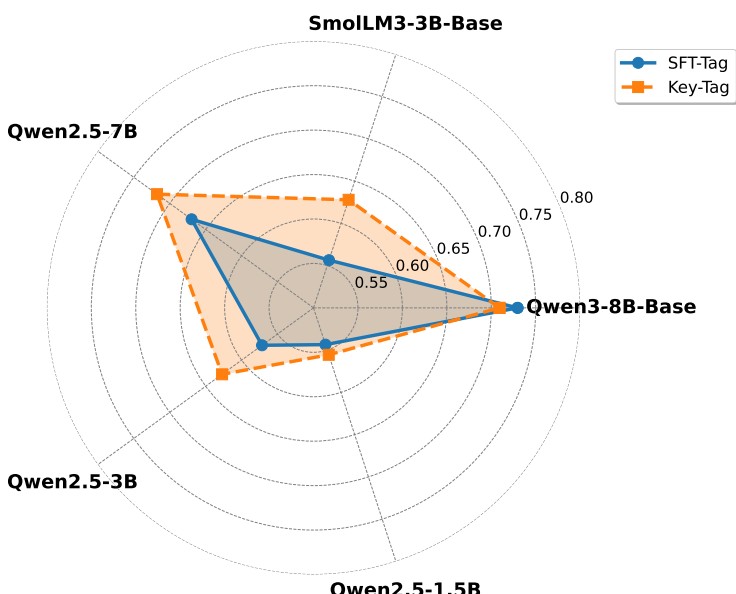

Figure 2: Boxplot showing the distribution of accuracy differences between Key-Tag and SFT-Tag for individual models across multiple datasets. Each of the first five boxes represents a single model's accuracy differences on four datasets, while the sixth box ("Avg-Acc") represents the distribution of average accuracy differences across all models.

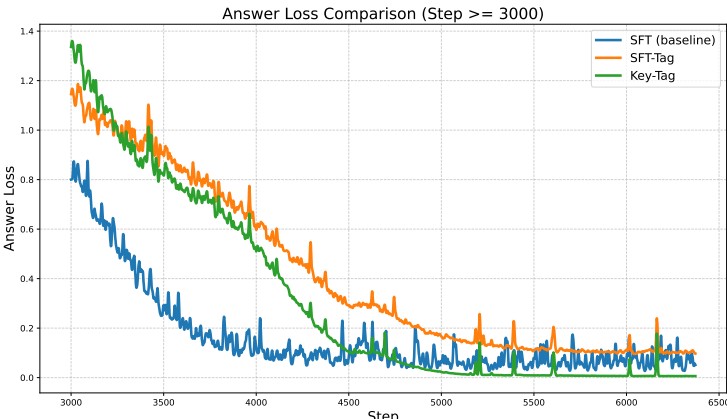

Figure 3: Comparison of answer-level loss on GSM8K for Qwen2.5-7B under different training strategies. The plot shows the loss curves for baseline SFT, SFT-Tag, and Key-Tag, highlighting the effect of structured tagging and key-focused optimization on the model's convergence and answer accuracy.

This observation is consistent with the trend of the Key loss curves shown in Figure 3. At the early stages of training, the inclusion of structure tags causes Key-Tag to exhibit a higher Key loss compared to SFT. However, as training progresses, the Key-Tag strategy gradually guides the model to better focus on the answer itself, leading to a continual decrease in Answer loss. Ultimately, the Answer loss of Key-Tag falls below that of SFT, further confirming its advantage in enhancing answer accuracy.

Although Key-Tag improves accuracy, this gain comes at the cost of output structure. Supervised fine-tuning (SFT-Tag) treats all tokens uniformly without assigning special emphasis to the answer. As a result, although this approach may yield slightly lower accuracy, it is more effective in capturing

the structural organization of responses. When evaluating the model's ability to correctly generate the designed tags `<Thinking></Thinking>` and `<Answer></Answer>`, Key-Tag exhibits poorer formatting and reduced readability, whereas SFT-Tag maintains well-structured outputs, as shown in Table 3.

| Model | Method | GSM8K | OpenR1-Math-220k | OpenBookQA | CoT-Collection | AvgFmt |
|---|---|---|---|---|---|---|
| Qwen3-8B-Base | SFT-Tag | 0.9984 | 0.8884 | **1.0000** | **1.0000** | 0.9717 |
|  | Key-Tag | 0.5910 | 0.5929 | 0.9072 | 0.9138 | 0.7512 |
| Qwen2.5-7B | SFT-Tag | **0.9984** | 0.8460 | **1.0000** | **0.9960** | 0.9601 |
|  | Key-Tag | 0.0000 | 0.0000 | 0.0000 | 0.0000 | 0.0000 |
| SmolLM3-3B-base | SFT-Tag | 0.9924 | **0.9032** | 0.9360 | 0.9939 | **0.9564** |
|  | Key-Tag | 0.0007 | 0.0091 | 0.2580 | 0.0040 | 0.0679 |

Table 3: Format adherence comparison between the Key-Tag and SFT-Tag training methods across various models. The last column reports the average format adherence for each model, providing a comprehensive summary of overall performance. For each dataset, the best-performing results for a given model are highlighted in bold to emphasize optimal performance.

Overall, the ablation results highlight a clear trade-off between accuracy and output structure. While the Key-Tag approach consistently improves answer correctness across different models, this comes at the cost of reduced adherence to the desired output format. In contrast, SFT-Tag treats every token equally, which helps maintain proper output formatting, resulting in more structured and reliable responses.

## 5.3 ABLATION STUDY ON ONE-STAGE VS. TWO-STAGE TRAINING STRATEGIES

| Model | Method | GSM8K | OpenR1-Math-220k | OpenBookQA | CoT-Collection | Avg_Acc |
|---|---|---|---|---|---|---|
| Qwen3-8B-Base | SFT-Tag | 0.8300 | 0.6134 | **0.8660** | 0.6092 | 0.7297 |
|  | Key-Tag | 0.7982 | 0.5789 | 0.8629 | **0.5972** | 0.7093 |
|  | **SFTKey-Tag(ours)** | **0.8309** | **0.8116** | 0.8600 | 0.6140 | **0.7791** |
| Qwen2.5-7B | SFT-Tag | 0.7582 | 0.4380 | 0.8560 | 0.6260 | 0.6696 |
|  | Key-Tag | **0.7885** | 0.5460 | 0.896 | **0.6400** | **0.7176** |
|  | **SFTKey-Tag(ours)** | 0.7809 | **0.6529** | **0.8920** | 0.6220 | 0.7369 |
| SmolLM3-3B-Base | SFT-Tag | 0.6854 | 0.4194 | 0.6180 | 0.5030 | 0.5565 |
|  | Key-Tag | **0.8354** | 0.3761 | **0.7740** | 0.5260 | **0.6278** |
|  | **SFTKey-Tag(ours)** | 0.6884 | **0.4677** | 0.7455 | **0.5442** | 0.6114 |

Table 4: Accuracy performance of different models trained with one-stage strategy (SFT-Tag and Key-Tag) and the two-stage strategy (SFTKey-Tag) across multiple datasets. The last column reports the average accuracy across datasets for each model.

| Model | Method | GSM8K | OpenR1-Math-220k | OpenBookQA | CoT-Collection | Avg-Fmt |
|---|---|---|---|---|---|---|
| Qwen3-8B-Base | SFT-Tag | 0.9984 | 0.8884 | **1.0000** | **1.0000** | 0.9717 |
|  | Key-Tag | 0.5910 | 0.5929 | 0.9072 | 0.9138 | 0.7512 |
|  | **SFTKey-Tag(ours)** | **1.0000** | **0.9839** | **1.0000** | **1.0000** | **0.9959** |
| Qwen2.5-7B | SFT-Tag | **0.9984** | 0.8460 | **1.0000** | **0.9960** | 0.9601 |
|  | Key-Tag | 0.0000 | 0.0000 | 0.0000 | 0.0000 | 0.0000 |
|  | **SFTKey-Tag(ours)** | 0.9848 | **0.8823** | 0.9900 | **0.9960** | **0.9632** |
| SmolLM3-3B-base | SFT-Tag | 0.9924 | **0.9032** | 0.9360 | 0.9939 | **0.9564** |
|  | Key-Tag | 0.0007 | 0.0091 | 0.2580 | 0.0040 | 0.0679 |
|  | **SFTKey-Tag(ours)** | **0.9931** | 0.6505 | **0.9939** | **0.9959** | 0.9084 |

Table 5: Format performance of different models trained with single-stage strategy (SFT-Tag and Key-Tag) and the two-stage strategy (SFTKey-Tag) across multiple datasets. For each dataset within the same model, the highest value is highlighted in bold. The last column reports the average format adherence (Avg-Fmt) for each model.

As shown in Table 4,With the addition of the Key-tag, the two-stage SFTKey-Tag outperforms the one-stage SFT-Tag on most benchmarks, achieving an average accuracy improvement of 5%. Additionally, As presented in Table 5,we compare the format adherence between one-stage and two-stage strategy. The SFTKey-Tag maintains significantly better output formatting compared to Key-Tag. This highlights the critical role of the SFT-Tag stage in learning and maintaining proper output formatting.

In summary, these observations indicate that both stages in SFTKey-Tag are necessary. The two-stage approach effectively combines the strengths of each single-stage method while mitigating their respective weaknesses, thereby validating the effectiveness of the SFTKey-Tag training strategy.

## 6 RELATED WORKS

**Token Importance.** In supervised fine-tuning (SFT), the relative importance of individual tokens plays a crucial role in shaping model performance (Sow et al., 2025; Pang et al., 2025; Zhang et al., 2025). Not all tokens contribute equally to learning, and prioritizing critical or informative tokens can guide the model more effectively. For instance, Lin et al. (2024) emphasize identifying and focusing on the most informative tokens, showing that proper token weighting—whether based on importance, positional range, or informativeness—can substantially affect fine-tuning outcomes.

**Token Importance based SFT.** Beyond conventional SFT, several methods have been proposed to incorporate token importance into SFT. Kim et al. (2025) introduce importance groups in SFT-Go, assigning higher weights to critical token groups to emphasize essential parts of the output. Similarly, Helm et al. (2025) distinguish between short-range and long-range tokens, adjusting their contributions to improve learning over extended contexts. While these approaches enhance performance, they generally rely on predefined notions of token importance, introducing extra hyperparameters and making training sensitive to dataset-specific characteristics, which can limit generalization across tasks or domains.

## 7 CONCLUSION

In this work, we propose SFTKey-Tag, a two-stage fine-tuning strategy that balances output formatting and answer correctness. The first stage (SFT-Tag) ensures well-structured outputs, while the second stage (Key-Tag) strengthens accuracy by focusing on final answers. Our theoretical analysis disentangles the contributions of these two stages, showing that their integration achieves complementary improvements.

Through empirical studies, we first observe that introducing tags in SFT can change accuracy across datasets. Separately, we design the Key-Tag method, which improves answer correctness by focusing solely on the Key portion, but at the same time introduces inconsistencies in output formatting. To address this trade-off, SFTKey-Tag combines the strengths of two stages. Experiments across multiple models and benchmarks demonstrate that SFTKey-Tag consistently outperforms the baseline SFT approach, delivering more accurate predictions while preserving coherent and reliable outputs.

## 8 LIMITATION

Due to computational and experimental resource constraints, we did not extend our evaluation to larger-scale models (e.g., 14B, 32B) or to a broader range of benchmarks. Our study focused on models with 1.5B, 3B, 7B, and 8B parameters. Moreover, since the proposed SFTKey-Tag introduces an additional key-focused training stage, each experiment required longer training time compared to conventional SFT. Consequently, our evaluation was limited to general-domain and mathematical datasets, and future work should explore additional domains.

## 9 ETHICS STATEMENT

Our work focuses on exploring novel supervised fine-tuning (SFT) methods for large language models. All datasets and models used are publicly available and do not contain personally identifiable information. We acknowledge that large language models may reflect societal biases present in the training data. We encourage responsible use of these models and advise against deploying them in high-stakes applications without further assessment.

## 10 REPRODUCIBILITY STATEMENT

We use publicly available datasets and models, and provide comprehensive details of the training procedures and hyperparameters to ensure that our experiments can be fully reproduced.

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

## A  USE OF LLMS

We mainly employ large language models (LLMs) for training and evaluation, using them to generate predictions, perform reasoning, and assess model performance under various settings. The specific prompts used in these experiments are shown below. Additionally, we leverage LLMs to assist in optimizing and refining the content of this paper, including improving clarity, readability, and overall presentation.

## B  COMPUTE RESOURCES FOR EXPERIMENTS

All experiments were conducted using 8 NVIDIA A100 GPUs (40GB each). On average, training a single model required approximately 2 hours for Qwen2.5-7B and Qwen3-8B, 1.5 hours for Qwen2.5-3B and SmolLM3-3B-Base, and 1 hour for Qwen2.5-1.5B. The total GPU consumption for the experiments reported in this paper is estimated to be within 1,000 GPU hours. Considering additional exploratory runs and extended experiments beyond those presented in the main text, the overall computational budget does not exceed 2,000 GPU hours.

## C  PROMPT

### C.1  TRAINING PROMPT

```
A conversation between User and Assistant. The user asks a question,and
    the Assistant solves it.The assistant first thinks about the
    reasoning process in the mind and then provides the user with the
    answer.The reasoning process and answer are enclosed within
    <Thinking> </Thinking> and <Answer> </Answer> tags, respectively,
    i.e.,<Thinking> reasoning process here </Thinking><Answer> answer
    here </Answer>.
```

### C.2  EVAL PROMPT

```
You are a expert. Please determine whether the answers in the two
    responses to the following question are the same. If they are the
    same, reply with yes; if they are different, reply with no.
Note:
You do not need to provide any a n a l y s i s just reply with yes or no.

**Question**:
{question}
**Response 1**:
{answer1}

**Response 2**:
{answer2}

Please make your judgment and provide your reply.
```

## D  FORMAT EVALUATION

Since we define the <Thinking></Thinking> and <Answer></Answer> tags in the model outputs, we check whether the generated content conforms to the specific format <Thinking>Reasoning Content</Thinking><Answer>Answer Content</Answer>. Based on this, we compute the proportion of outputs that follow this format, thereby quantifying the accuracy of the output formatting.

# E   SUMMARY OF MODEL ACCURACY

| Model | Method | GSM8K | OpenR1-Math-220k | OpenBookQA | CoT-Collection | Avg_Acc |
|---|---|---|---|---|---|---|
| Qwen3-8B-Base | SFT(baseline) | **0.8378** | 0.5180 | 0.8600 | 0.6120 | 0.7069 |
| | SFT-Tag | 0.8300 | 0.6134 | **0.8660** | 0.6092 | 0.7297 |
| | Key-Tag | 0.7982 | 0.5789 | 0.8629 | **0.7093** | 0.7512 |
| | **SFTKey-Tag**(ours) | 0.8309 | **0.8116** | 0.8600 | 0.6140 | **0.7791(+0.0722)** |
| SmolLM3-3B-Base | SFT(baseline) | 0.6732 | 0.3140 | **0.7880** | 0.5280 | 0.5758 |
| | SFT-Tag | 0.6854 | 0.4194 | 0.6180 | 0.5030 | 0.5565 |
| | Key-Tag | **0.8354** | 0.3761 | 0.774 | 0.526 | **0.6278** |
| | **SFTKey-Tag**(ours) | 0.6884 | **0.4677** | 0.7455 | **0.5442** | 0.6115(+0.0357) |
| Qwen2.5-7B | SFT(baseline) | 0.7589 | 0.4620 | 0.8460 | 0.6220 | 0.6722 |
| | SFT-Tag | 0.7582 | 0.4380 | 0.8560 | 0.6260 | 0.6696 |
| | Key-Tag | **0.7885** | 0.546 | 0.896 | **0.64** | **0.7176** |
| | **SFTKey-Tag**(ours) | 0.7809 | **0.6529** | **0.8920** | 0.6220 | 0.7369(+0.0647) |
| Qwen2.5-3B | SFT(baseline) | 0.6785 | 0.4420 | 0.7340 | 0.5320 | 0.5966 |
| | SFT-Tag | 0.5524 | 0.4860 | 0.7120 | 0.5360 | 0.5719 |
| | Key-Tag | **0.7187** | **0.5100** | **0.7900** | 0.49 | **0.6271** |
| | **SFTKey-Tag**(ours) | 0.6770 | 0.4880 | 0.7100 | **0.5711** | 0.6115(+0.0104) |
| Qwen2.5-1.5B | SFT(baseline) | 0.4640 | 0.4060 | 0.6480 | 0.4640 | 0.4955 |
| | SFT-Tag | 0.4760 | 0.4000 | 0.8220 | 0.4760 | 0.5435 |
| | Key-Tag | 0.49 | 0.4040 | **0.8380** | **0.49** | 0.5555 |
| | **SFTKey-Tag**(ours) | **0.5269** | **0.4906** | 0.7440 | 0.4560 | 0.5543(+0.0548) |

Table 6: Accuracy performance of different models trained with various strategies across multiple datasets. The last column reports the average accuracy across datasets for each model. Within each dataset, the highest accuracy achieved under different training strategies for the same model is highlighted in bold to indicate the best result. Relative improvements over the baseline SFT are shown in parentheses to emphasize performance gains.

# F   SUMMARY OF MODEL FORMAT

| Model | Method | GSM8K | OpenR1-Math-220k | OpenBookQA | CoT-Collection | Avg-Fmt |
|---|---|---|---|---|---|---|
| Qwen3-8B-Base | SFT | 0.7946 | 0.8460 | **1.0000** | 0.9980 | 0.9071 |
| | SFT-Tag | 0.9984 | 0.8884 | **1.0000** | **1.0000** | 0.9717 |
| | Key-Tag | 0.5910 | 0.5929 | 0.9072 | 0.9138 | 0.7512 |
| | **SFTKey-Tag** | **1.0000** | **0.9839** | **1.0000** | **1.0000** | **0.9959** |
| SmolLM3-3B-base | SFT | 0.9909 | 0.7920 | 1.0000 | 0.9940 | 0.9442 |
| | SFT-Tag | 0.9924 | **0.9032** | 0.9360 | 0.9939 | **0.9564** |
| | Key-Tag | 0.0007 | 0.0091 | 0.2580 | 0.0040 | 0.0679 |
| | **SFTKey-Tag** | **0.9931** | 0.6505 | **0.9939** | **0.9959** | 0.9084 |
| Qwen2.5-7B | SFT | 0.9977 | 0.8460 | **1.0000** | **0.9980** | 0.9604 |
| | SFT-Tag | **0.9984** | 0.8460 | **1.0000** | 0.9960 | 0.9601 |
| | Key-Tag | 0.0000 | 0.0000 | 0.0000 | 0.0000 | 0.0000 |
| | **SFTKey-Tag** | 0.9848 | **0.8823** | 0.9900 | 0.9960 | **0.9632** |

Table 7: Format performance of different models trained with SFT, SFT-Tag, Key-Tag, and SFTKey-Tag strategies across multiple datasets. For each dataset within the same model, the highest value is highlighted in bold. The last column reports the average format adherence (Avg-Fmt) for each model.