# OpenReview forum: "Rethinking Supervised Fine-Tuning: Emphasizing Key Answer Tokens for Improved LLM Accuracy"
_ICLR.cc/2026/Conference — Submitted to ICLR 2026_

### Official Review · Reviewer_R2DL · 2025-10-25

**Soundness:** 3
**Presentation:** 2
**Contribution:** 2
**Rating:** 2
**Confidence:** 4

**Summary:**

This Paper proposes a two-stage fine-tuning strategy, SFTKey-Tag, to address the issue in SFT where LLMs excessively focus on CoT and overlook the key answer components. In the first stage, the authors conduct SFT with special tokens to ensure the format is correct. In the second stage, only the tokens of the answer part are fine-tuned for improving the accuracy. Experiments are conducted on five models and four datasets. The results shows that the proposed method outperforms the conventional SFT method by over 5 percent in average accuracy, while maintaining the integrity of the output format.

**Strengths:**

The definition of the research problem is clear and well focused, addressing the issue where LLMs tend to focus on CoT rather than the answer part. The method proposed is innovative and relatively simple and the experimental results support the effectiveness of the method.

**Weaknesses:**

1. The experiments in this paper only study models with 1.5B-8B parameters, excluding larger models (e.g. 14B, 32B) that are widely used in practice, and the selected models are dominated by Qwen series (4 out of 5) and SmolLM3, with no coverage of other mainstream architectures (e.g. Llama, Mistral).
2. While the paper mentions that SFTKey-Tag requires longer training time than conventional SFT, it provides no quantitative data on the time overhead or potential optimization strategies.
3. Ablation results show that tagging (SFT-Tag) has inconsistent effects on accuracy across datasets, but the paper does not explore the underlying causes. Without a clear mechanism, users cannot determine when tagging is beneficial for their specific tasks, reducing the method’s usability.
4. While the paper emphasizes the importance of CoT for complex reasoning, it only evaluates the accuracy of the final answer and output format, ignoring the quality of the CoT itself. The study fails to fully validate whether SFTKey-Tag perserves or degrades the reasoning ability of LLMs, even if the final answer is correct.

**Questions:**

1.How would the proposed SFTKey-Tag perform on these larger or more diverse models? Could the authors discuss potential scalability or generalization issues?
2.Could the authors provide timing benchmarks or suggest potential optimization strategies to mitigate this overhead? Is the performance boost worth the trade-off in training time and complexity?
3.Could the authors provide insights into when and why tagging is beneficial or detrimental for specific tasks? Is the CoT consistent with the key answer?
4.How does SFTKey-Tag affect the reasoning process of LLMs? Could the authors include metrics or analyses to verify whether the method preserves or degrades reasoning capabilities? Does the method affect the LLM’s performance or output length?
5.Could the authors compare the SFTKey-Tag method with the baselines of other related methods?(e.g. https://arxiv.org/pdf/2404.07965, https://arxiv.org/pdf/2508.05629, https://arxiv.org/pdf/2506.12860, https://arxiv.org/pdf/2503.02875)
6.Could the authors test the performance of this method on other more complex datasets? (e.g. AIME 2024, AIME2025)

---

### Official Review · Reviewer_t1HN · 2025-10-31

**Soundness:** 2
**Presentation:** 1
**Contribution:** 2
**Rating:** 2
**Confidence:** 4

**Summary:**

This paper introduces SFTKey, a two-stage supervised fine-tuning framework for large language models (LLMs). The core idea is to separate model outputs into reasoning and answer segments using special tags `<Thinking>` and `<Answer>`. In the first stage, standard SFT is applied to train both reasoning and answer portions to ensure output format consistency. In the second stage, optimization is restricted to the answer segment to enhance answer accuracy. Experiments across several benchmarks (e.g., GSM8K, OpenR1-Math-220K, OpenBookQA, CoT-Collection) and multiple model scales show that the proposed SFTKey-Tag method improves composite accuracy scores by approximately 5% over the conventional SFT baseline. The idea of selectively emphasizing key tokens during fine-tuning is interesting. However, several methodological and presentation issues limit the paper’s clarity, rigor, and overall contribution.

**Strengths:**

The paper focuses on an important problem in fine-tuning LLMs for reasoning: the imbalance between reasoning and answer tokens.
The proposed two-stage SFTKey approach is conceptually simple and easy to implement.
The empirical results show moderate improvements in composite accuracy over standard SFT.

**Weaknesses:**

1. Incorrect figure labeling: In Figure 1, the distinction between “Training” and “Loss Computation” is misleading. The figure should illustrate whether *loss* is applied to each token rather than whether the token participates in training.
2. Dataset clarity: Line 164 only refers vaguely to benchmarks without specifying the actual training–validation partitions or whether test data were held out. This undermines reproducibility.
3. Limited novelty: The use of `<Thinking>` and `<Answer>` tags to separate reasoning and answers has been adopted in prior “thinking LLM” works (e.g., DeepSeek-R1 2025), so this cannot be considered a novel contribution.
4. Poor formatting: The manuscript contains large blank spaces on pages 4, 6, and 7, which detracts from readability.
5. Typo: There is an extraneous bracket [ at line 260 and several minor stylistic inconsistencies throughout.
6. Unconvincing evaluation metric: Table 1 reports a “Score” metric defined as a 70% accuracy + 30% format accuracy. This weighting appears arbitrary and tailored to highlight the proposed method. The claim of achieving “SOTA” based on this metric is therefore not credible.
7. Figure mislabeling: Figure 2 is titled as a Boxplot but is actually a Radar Plot, which should be corrected.
8. Possible overfitting: As shown in Figure 3, the proposed method’s loss curve indicates potential overfitting. The authors should validate on larger and more diverse benchmarks to ensure robustness.
9. Insufficient analysis: The paper lacks deeper analysis on why answer-only fine-tuning improves performance and under what conditions it fails. No ablation is presented for tag effectiveness beyond small-scale comparisons.

**Questions:**

1. How were the datasets split between training, validation, and testing? Are any benchmark test sets used for training?
2. Why was the composite metric weighted as 0.7 × accuracy + 0.3 × format? Have alternative weighting ratios been tested?
3. Can the authors provide standard deviations or statistical significance tests to support the claimed 5% gain?
4. Have you tested whether the second fine-tuning stage degrades reasoning ability or coherence of intermediate steps?
5. Could SFTKey-Tag generalize to open-ended generation tasks (e.g., code or summarization) rather than structured reasoning datasets?

---

### Official Review · Reviewer_1jyq · 2025-11-01

**Soundness:** 1
**Presentation:** 2
**Contribution:** 1
**Rating:** 0
**Confidence:** 4

**Summary:**

The paper proposes a two-stage SFT strategy: first fine-tuning on the full model responses (including the reasoning chain), followed by a second stage that focuses exclusively on fine-tuning the answer portion within those responses.

**Strengths:**

This paper is clear and easy to follow.

**Weaknesses:**

1. Concerns about the methodological soundness: The method lacks a compelling rationale, and the authors provide neither theoretical analysis nor empirical justification for its design. The reported results are insufficient to demonstrate the method’s effectiveness, as the observed gains could stem from various confounding factors—such as under-training in the first stage or overfitting in the second stage—rather than the proposed two-stage scheme itself.
2. Limited evaluation on simplistic benchmarks: The datasets chosen for evaluation are relatively simple, and the final model performance remains low—for instance, achieving only around 80% on GSM8K, which is notably subpar by current standards. Consequently, the experiments fail to establish whether the method generalizes to stronger models or more challenging, real-world reasoning tasks.
3. Lack of motivation and deeper analysis: As noted above, the authors offer no theoretical motivation or insightful empirical analysis. While the paper examines the impact of special tokens on reasoning outcomes, it overlooks a substantial body of relevant work beyond the cited references. For example, recent approaches such as [1] provide valuable frameworks for analyzing token-level effects in reasoning processes. The authors should conduct a more thorough and nuanced investigation to substantiate their claims.

[1] Enhancing Chain-of-Thought Reasoning with Critical Representation Fine-tuning ACL 2025

**Questions:**

Please see the weaknesses.

---

### Official Review · Reviewer_yfaf · 2025-11-02

**Soundness:** 3
**Presentation:** 3
**Contribution:** 2
**Rating:** 2
**Confidence:** 4

**Summary:**

This paper introduces a two-stage supervised fine-tuning (SFT) method named SFTKey-Tag. It aims to address the issue of large language models (LLMs) inadequately focusing on the accuracy of the final answer when generating responses that include a chain-of-thought (CoT). The method first involves standard fine-tuning on data labeled with `<Thinking>` and `<Answer>` tags to learn the output format. Subsequently, in the second stage, the loss is calculated and training is performed exclusively on the key answer part within the `<Answer>` tag.

**Strengths:**

1.  Clear Objective and Loss Definition: The paper astutely identifies a key deficiency in standard supervised fine-tuning (SFT) for Chain-of-Thought (CoT) tasks: the disproportionate allocation of model loss between the lengthy reasoning steps and the concise final answer, which may lead to insufficient optimization of the final answer's accuracy. This problem is defined with great clarity, providing a precise target for the proposed method.
2.  Simplicity and Practicality: The SFTKey-Tag method is remarkably simple and intuitive, adding minimal implementation complexity or hyperparameter tuning overhead. This makes it highly practical for real-world application.
3.  Comprehensive Experimental Design:The study covers multiple models and datasets, with thorough ablation studies.

**Weaknesses:**

1.  Lack of Novelty and Insufficient Literature Review: The use of "structured labels + fine-tuning" is already an active research area for enhancing model reasoning capabilities, with various implementation paths being explored. However, the paper fails to provide a sufficient comparison or discussion with recent alternative approaches that employ more complex labeling schemes or integrate reinforcement learning (e.g., arXiv:2506.20241). This omission makes it difficult to ascertain the novelty and state-of-the-art standing of the proposed method.
2.  Lack of Comparison with Mainstream Methods: The paper only compares its method against its own baselines, without benchmarking against any current state-of-the-art (SOTA) methods. This prevents the validation of its claimed advancements.
3.  This approach to evaluation is methodologically questionable. The use of a composite metric with pre-defined, arbitrary weights (0.7 and 0.3) without sufficient justification raises concerns about its objectivity. It is possible that this specific weighting inadvertently favors the performance profile of the proposed method.

**Questions:**

1.  The paper's core claim is the enhancement of LLM accuracy, yet all experiments are conducted between internal baselines. Without any benchmarking against publicly established State-of-the-Art (SOTA) models, how can the true value and competitiveness of the SFTKey-Tag method be objectively assessed?
2.  The paper provides an operational procedure but fails to explain the underlying principles. Why is the second stage of SFTKey-Tag effective in improving answer accuracy while not causing "catastrophic forgetting" of the formatted output capabilities learned in the first stage? Is this merely a heuristically effective engineering trick, or is there a deeper learning dynamic or theoretical basis that can explain this phenomenon?

---

### Meta-Review · Area_Chair_3Tjk · 2026-01-12

**Summary:**

This paper proposes SFTKey-Tag, a two-stage supervised fine-tuning (SFT) method that first trains LLMs on full chain-of-thought responses tagged with <Thinking> and <Answer>, then refines the model by applying loss only to the answer segment to improve final-answer accuracy. Experiments across multiple benchmarks and model scales report ~5% gains in a composite accuracy metric over standard SFT. However, the work suffers from several limitations. The core idea—structured tagging combined with selective fine-tuning—is not sufficiently differentiated from existing approaches, and the paper lacks a thorough discussion or comparison with recent related methods. Additionally, the evaluation relies on a composite metric with arbitrary weights (0.7/0.3) that are not justified.

**Reviewer Scores:**

NA

---

### Decision · Program_Chairs · 2026-01-26

Reject